# First Report on Natural Infection of Nodavirus in an Echinodermata, Sea Cucumber (*Apostichopus japonicas*)

**DOI:** 10.3390/v13040636

**Published:** 2021-04-08

**Authors:** Chong Wang, Liang Yao, Wei Wang, Songwen Sang, Jingwei Hao, Chenghua Li, Qingli Zhang

**Affiliations:** 1School of Marine Sciences, Ningbo University, Ningbo 315311, China; wangchongyilin@163.com; 2Yellow Sea Fisheries Research Institute, Chinese Academy of Fishery Sciences, Function Laboratory for Marine Fisheries Science and Food Production Processes (Qingdao), National Laboratory for Marine Science and Technology, Key Laboratory of Maricultural Organism Disease Control, Ministry of Agriculture, Qingdao Key Laboratory of Mariculture Epidemiology and Biosecurity, Qingdao 266071, China; yaoliang2019@163.com (L.Y.); 13173113773@163.com (W.W.); sangxihang@163.com (S.S.); 15764228381@163.com (J.H.)

**Keywords:** covert mortality nodavirus (CMNV), natural infection, sea cucumber (*Apostichopus japonicas*), *in situ* hybridization, TEM

## Abstract

Cross-species transmission of emerging viruses happens occasionally due to epidemiological, biological, and ecological factors, and it has caused more concern recently. Covert mortality nodavirus (CMNV) was revealed to be a unique shrimp virus that could cross species barrier to infect vertebrate fish. In the present study, CMNV reverse transcription-nested PCR (RT-nPCR)-positive samples were identified from farmed sea cucumber (*Apostichopus japonicas*) in the CMNV host range investigation. The amplicons of RT-nPCR from sea cucumber were sequenced, and its sequences showed 100% identity with the RNA-dependent RNA polymerase gene of the original CMNV isolate. Histopathological analysis revealed pathologic changes, including karyopyknosis and vacuolation of the epithelial cells, in the sea cucumber intestinal tissue. The extensive positive hybridization signals with CMNV probe were shown in the damaged epithelial cells in the *in situ* hybridization assay. Meanwhile, transmission electron microscopy analysis revealed CMNV-like virus particles in the intestine epithelium. All the results indicated that the sea cucumber, an Echinodermata, is a new host of CMNV. This study supplied further evidence of the wide host range of CMNV and also reminded us to pay close attention to its potential risk to threaten different aquaculture animal species.

## 1. Introduction

For most viruses, the virus–receptor interactions determine viral host range and therefore constitute the interspecies barrier of viral infection, eventually leading to the strong host specificity of a virus [1,2]. However, the cross-species transmission of emerging viruses happens occasionally due to a variety of epidemiological, biological, and ecological factors [3,4,5]. Furthermore, RNA viruses more easily cross species boundaries on account of the lack of exonuclease proofreading activity and easy variation [6], such as coronaviruses [7], the avian influenza virus [8], and the rabies virus [9].

*Nodaviridae* is composed of two genera, *Alphanodavirus* and *Betanodavirus* [10]. *Alphanodaviruses* were mostly isolated from insects and their host range also appears to be restricted to insects [11], except for Nodamura virus (NoV) and Flock House virus (FHV) [12,13]. NoV was originally isolated from mosquitoes (*Culex tritaeniorhynchus*) [14]; however, it could also lethally infect mammals, including suckling mice and suckling hamsters [10,15]. FHV, another alphanodavirus isolated from *Costelytra zealandica* (*Coleoptera*: *Scarabaeidae*), is capable of replicating in many species of plants, including chenopodium *Chenopodium hybridum*, barley *Hordeum vulgare*, and tobacco *Nicotiana tabacum* [12,16]. Therefore, the hosts of NoV and FHV are not only confined to insects, and they also possess the capacity of cross-species transmission as well. However, *Betanodavirus* mainly infects larvae, juvenile or adult marine fish, and is different from *Alphanodavirus* [17,18,19].

Covert mortality nodavirus (CMNV) is an alphanodavirus first isolated from shrimp with viral covert mortality disease (VCMD) in China and later also found in Thailand and Ecuador [20,21,22,23]. It can also infect various major farmed shrimp, including *Penaeus vannamei*, *Penaeus chinensis*, *Marsupenaeus japonicus*, and *Penaeus monodon*, and VCMD outbreaks in shrimp farms represent a significant threat to the shrimp culture industry [21]. Additionally, other crustaceans (including the hyperiid amphipod *Parathemisto gaudichaud*, amphipod *Corophium sinense Zhang*, and the ghost crab *Ocypode cordimandus*) and several fish species (including gobiid fish *Mugilogobius abei*, Japanese flounder *Paralichthys olivaceus*, and goldfish *Carassius auratus*) are also natural hosts of CMNV. And the host range spanning crustaceans and fish illustrates the capacity of CMNV to spread crossing the species barrier [24,25,26].

Sea cucumber (Holothuians) is a common marine invertebrate [27], and it is the generic term for *Holothuroidea*, which belongs to the invertebrate *Echinodermata*. *Apostichopus japonicus* is the sea cucumber species with the largest social demand, and its aquaculture has become an emerging marine industry [28,29,30]. In a systematic investigation of CMNV natural hosts and vectors, CMNV reverse transcription-nested PCR (RT-nPCR)-positive sea cucumber individuals were accidentally found in shrimp farming ponds. In this study, we describe here the outcome of the detection of CMNV in sea cucumber by RT-nPCR, *in situ* hybridization (ISH), histopathology, and transmission electron microscopy (TEM). Our study provides significant novel insights into the new natural host discovery of CMNV.

## 2. Materials and Methods

### 2.1. Sample Collection

In June 2018, we found that the farming *P. vannamei* collected from polyculture ponds of shrimp and sea cucumber was infected with CMNV. Considering the capacity of CMNV cross-species infection and the needs of further exploration of its host range, four sea cucumber individuals (length 7–8 cm, co-inhabiting with the *P. vannamei* in the same polyculture pond) (Figure 1a,b) were randomly collected for CMNV detection. The collected sea cucumber individuals looked normal, but the body was not compacted enough (somewhat soft) compared with healthy individuals. These sea cucumbers were primarily examined after dissecting the body along the longitudinal axis, and the thinning intestinal tissue (Figure 1c) was selected to be divided into three parts and preserved. One part was preserved in 4% paraformaldehyde solution in PBS (PFA-PBS) (Sinopharm, Beijing, China) for ISH detection and histopathological analysis. Another part was fixed in 2.5% glutaraldehyde solution (Solarbio, Beijing, China) for electron microscopic examinations. Residual intestinal tissues were preserved in RNAstore solution (Tiangen, Beijing, China) for molecular biological analysis.

### 2.2. Total RNA Extraction

The total RNA of intestinal tissues was extracted from four sea cucumber individuals using RNAiso Plus Reagent (Takara, Dalian, China) according to the manufacturer’s instructions. The tissues samples were first homogenized in RNAiso Plus, and then trichloromethane was added into the homogenate for protein degeneration. Finally, isopropanol was used to get total RNA from liquid supernatant obtained by centrifugation. The concentration and purity of purified RNA were measured by Nanodrop 2000 (Thermo Scientific, Waltham, MA, USA).

### 2.3. Reverse Transcription-Nested PCR (RT-nPCR)

The total RNA from the intestinal tissue of sea cucumber was used as a template for RT-PCR analysis. First, the first-step PCR amplification was conducted by using the template of 1 μL total RNA (concentration of the template was 100–200 ng/μL) and a PrimeScript One Step RT-PCR Kit (TaKaRa, Dalian, China) with the primer sets of CMNV-F1/R1 (CMNV-F1: 5′-AAATACGGCGATGACG-3′, CMNV-R1: 5′-ACGAAGTGCCCA-CAGAC-3′) according to the recommended procedures, and the annealing temperature was 52 °C. Using the first-step RT-PCR products as templates, the second-step PCR was carried out using a TaKaRa Ex Taq Kit (TaKaRa, Dalian, China) with the primer sets of CMNV-D-F1/R1 (CMNV-D-F1: 5′-TCGCGTATTCGTGGAT-3′, CMNV-D-R1: 5′-TAGGGTCAAAAGGTGTAGT-3′), and the annealing temperature was 52 °C. The expected CMNV target fragments of the first and second rounds of the PCR amplifications were 619 bp and 413 bp amplicons from the CMNV RNA-dependent RNA polymerase (RdRp) gene, respectively. Then, the amplicons were resolved by 2% agarose gel electrophoresis for 0.5 h.

### 2.4. Sequence Alignment and Phylogenetic Tree Analysis

The amplicons (413 bp) from the second step of the RT-nPCR of sea cucumber were sent for commercial sequencing to Sangon Biological Engineering (Shanghai, China) Co. Ltd. The obtained 413 bp RdRp gene fragment sequences were subjected to multiple sequence alignment by the online software of BLASTn (https://www.ncbi.nlm.nih.gov/, accessed on 4 February 2021). Then, the phylogenetic tree, based on 25 relevant RdRp protein sequences retrieved from the GenBank database (Table 1) and the deduced amino acids sequence of the 413 bp gene fragment, was constructed by using the software MEGA 6.0 [31]. The tree was finally optimized through the online tool of iTOL (https://itol.embl.de/, accessed on 4 February 2021).

### 2.5. In Situ Hybridization (ISH) and Histopathological Analysis

Fixation, dehydration and paraffin embedding of the tissue samples were conducted following the histological method reported by Bell and Lightner [32]. Two paraffin-embedded sections (3 µm) were prepared. One of the sections was subjected to CMNV ISH analysis according to the published papers [21,25], and the other section was stained with routine hematoxylin and eosin-phloxine (H&E) according to previously described procedures [33]. The ISH sections were counterstained using the Nuclear Fast Red solution (Solarbio, Beijing, China) [34]. Finally, the sections of ISH detection and H&E staining were analyzed under the Nikon Eclipse E80i microscope (Nikon Co., Tokyo, Japan), and the image acquisition was accomplished through the slide scanning system of Pannoramic MIDI (3DHISTECH Ltd., Budapest, Hungary).

### 2.6. Transmission Electron Microscopy Analysis

To detect the presence of CMNV particles in sea cucumber, the intestinal tissues (approximately 1 mm^3^) were first preserved in 2.5% glutaraldehyde for 24 h at 4 °C, and then further fixed in 1% osmium tetroxide for 2 h. Finally, the tissues were embedded in plastic resin [35,36]. An ultramicrotome (Leica EM UC7) was used to prepare ultrathin sections (50 nm) of a resin block, and the obtained sections were stained with uranyl acetate and lead citrate [37,38]. Eventually, all the sections were examined using a JEOL JEM-1200 electron microscope (equipped with a field emission gun).

## 3. Results

### 3.1. Detection of CMNV in Sea Cucumber by RT-nPCR

Two out of the four RNA samples from the sea cucumbers produced the expected amplicons (413 bp) in the RT-nPCR assay (Figure 2a). In addition, the 413 bp amplicons of the second-step PCR were then sequenced to confirm the exact sequence information of RdRp-targeted fragments.

### 3.2. Phylogenetic Analyses

The result of multiple sequence alignment based on the 413 bp amplicons (413 bp, GenBank No. MW678771) from the second-step PCR showed that the CMNV sequences of sea cucumber shared 100% identity with the known CMNV RdRp gene (GenBank no. KM112247). Phylogenetic analysis revealed that the deduced amino acid sequences of CMNV RdRp from sea cucumber samples were clustered closely into the embranchment of the primordial CMNV isolate (Figure 2b). Additionally, the phylogenetic tree showed that the CMNV RdRp fragment sequences from sea cucumber were clustered into *Alphanodavirus*, and that members from *Betanodavirus* were clustered into the other independent branch (Table 1).

### 3.3. Detection of CMNV in Sea Cucumber by ISH and Histopathological Analysis

Purple positive hybridization signals of CMNV probe were widely present in the intestine of CMNV RT-nPCR-positive sea cucumber (Figure 3c), and especially more pronounced in intestinal epithelial cells (Figure 3d). Meanwhile, histological examination revealed obvious histopathological lesions in the same sites, such as villi decreasing, villus epithelium cells becoming uncompact and acidophilic (comparing with the normal villus epithelium cells), karyopyknosis (red arrows) and extensive vacuolation (yellow arrows) in the intestinal epithelial cells (Figure 3a,b). The result of ISH and histopathological analysis of the intestine of sea cucumber of negative control is shown in Figure 4a–d. No obvious histopathological changes and CMNV-positive hybridization signals were observed in the intestinal tissue (Figure 4c,d).

### 3.4. Detection of CMNV in Sea Cucumber by TEM Analysis

TEM analysis was used to further confirm the CMNV infection in the sea cucumber. TEM micrographs of the ultrathin sections of the intestinal tissue of sea cucumber revealed the presence of massive CMNV-like virus particles of about 28–32 nm in the intestinal villus epithelium cells (Figure 5a–d).

## 4. Discussion

Recently, the phenomenon of cross-species transmission of a few viruses has negatively affected areas such as human life, animal husbandry, and aquaculture and, meanwhile, also piqued the concern of scholars [25,39,40]. Notably, CMNV, an RNA virus isolated from invertebrate shrimp, can cross the host barrier to infect vertebrate fish, including the marine fishes gobiid fish *Mugilogobius abei* and Japanese flounder *Paralichthys olivaceus* [24,25], as well as the freshwater fish goldfish *Carassius auratus* [25]. CMNV infection in fish reminds that we should pay attention to the potential ability of CMNV to cross more species barriers. Hence, it makes sense to strengthen the investigation of the host range of CMNV. In this study, we proved that sea cucumber is a new natural host of CMNV—something that, to the best of our knowledge, has not been reported before in echinoderms.

CMNV-positive individuals were found in the farming sea cucumber by RT-nPCR. Thus, a further investigation was performed to confirm the infection of CMNV in sea cucumber. The histopathological analysis revealed that obvious pathological lesions, including karyopyknosis and vacuolation of the epithelial cells, occur in the sea cucumber intestinal tissue, and these lesions are similar to the intestinal lesions of gobiid fish *Mugilogobius abei* infected with CMNV [24]. Consistently, extensive positive hybridization signals of the CMNV probe were also revealed in the damaged epithelial cells in the ISH assay. Meanwhile, the presence of CMNV-like particles in lesion sites of the intestine was also confirmed by TEM analysis. In addition, it was found that the amplicons of RT-nPCR from sea cucumber were highly identical to the CMNV original RdRp gene. Therefore, the above results demonstrated that the sea cucumber could be infected by CMNV and turned into a new natural host of CMNV. In addition, sea cucumber has been continually attacked by disease in recent years [41,42,43], and whether CMNV is one of the important pathogens should be of concern.

During the past decade, the rapid and disorderly expansion of sea cucumber aquaculture was companied by various diseases that caused high mortality of sea cucumbers, resulting in grievous economic losses to the culture industry of sea cucumber [44,45]. Frequent outbreaks of disease has obviously hindered the sustainable development of sea cucumber culture industry. The studies on infectious agents of sea cucumber have been frequently reported in the past 10 years [46]. For many of these disease events, most research has focused on bacterial pathogens such as *Vibrio harveyi* [47], *Vibrio splendidus* [48], and *Lactococcus garviaeae* [49], while studies on viral pathogens have been relatively rare. Some scholars successfully isolated DNA viruses from diseased sea cucumbers, reproduced the disease symptoms, and speculated that the virus might be one of the pathogens causing diseases [46,50,51]. However, to date, no virus has been definitely identified and named. The discovery of CMNV in sea cucumber might provide more possibility and theoretical basis for the research of viral pathogens in diseased sea cucumber.

It is known that sea cucumber, a scavenger, is the pivotal component of the marine environment [52], playing an important role in the recycling of marine ecosystems [53]. In addition, sea cucumber is considered an important aquaculture species in East Asian countries because it has high nutritional value and is rich in multiple nutrients and biologically active substances [54,55,56,57]. Because of increased market demand, sea cucumber culture has become one of the pillar industries of China’s aquaculture industry [58,59]. The natural infection of CMNV in sea cucumber reveals the potential risk of CMNV to the maintaining of sea cucumber natural resources in marine ecosystems, as well as to the aquaculture of the increasingly important species.

## 5. Conclusions

Overall, the results of this study demonstrated that CMNV is a broad host range virus that deserves close attention to its potential risk to infect more aquaculture animal species. Considering the key ecological function of sea cucumber in the marine environment and the high economic value of this aquatic species, the potential negative impact of the spread and prevalence of CMNV in sea cucumber on the marine ecosystem and social economy should be of concern. The present study contributes to further investigation of the viral pathogen of diseased sea cucumber.

## Figures and Tables

**Figure 1 viruses-13-00636-f001:**
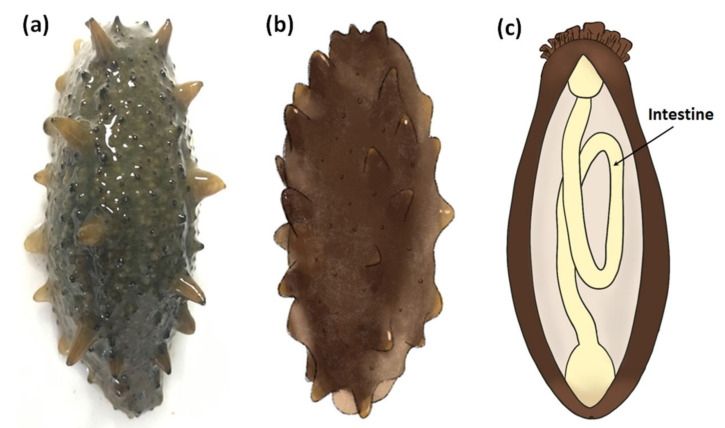
The topography and schematic diagram of the intestine of sea cucumber (*Apostichopus japonicas*) in this study. (**a**) Topography of sea cucumber collected from the polyculture ponds infected with covert mortality nodavirus (CMNV). (**b**) Schematic diagram of sea cucumber. (**c**) Schematic diagram of the longitudinal section of sea cucumber. Intestinal tissue is pointed by the black arrow.

**Figure 2 viruses-13-00636-f002:**
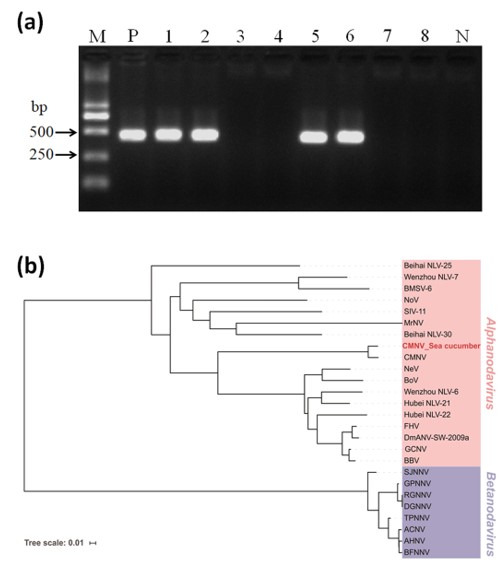
Electrophoretogram of amplicons from the CMNV RT-nPCR assay and phylogenetic analysis. (**a**) Electrophoretogram of amplicons from the second-step PCR of the CMNV RT-nPCR assay. M: DL2000 molecular weight marker. Lanes 1–8: the PCR products of four RNA samples from sea cucumbers’ intestines (each sample was done with two replicates; samples 1, 2, 3, and 4 were shown in lanes 1 and 2, 3 and 4, 5 and 6, 7 and 8, respectively). P: positive control; N: negative control. (**b**) Analysis of the phylogenetic tree based on the deduced amino acid sequences of the RdRp gene from the CMNV-positive sea cucumber sample and other nodaviruses (abbreviations of other viruses shown in Table 1). The CMNV isolate of the sea cucumber sampled from farming ponds is highlighted in red. Viral species of *Alphanodavirus* genus and *Betanodavirus* genus are shown in pink and lavender background, respectively. The phylogeny tree was derived using the neighbor-joining method by the MEGA 6.0 program. The scale bar is 0.01.

**Figure 3 viruses-13-00636-f003:**
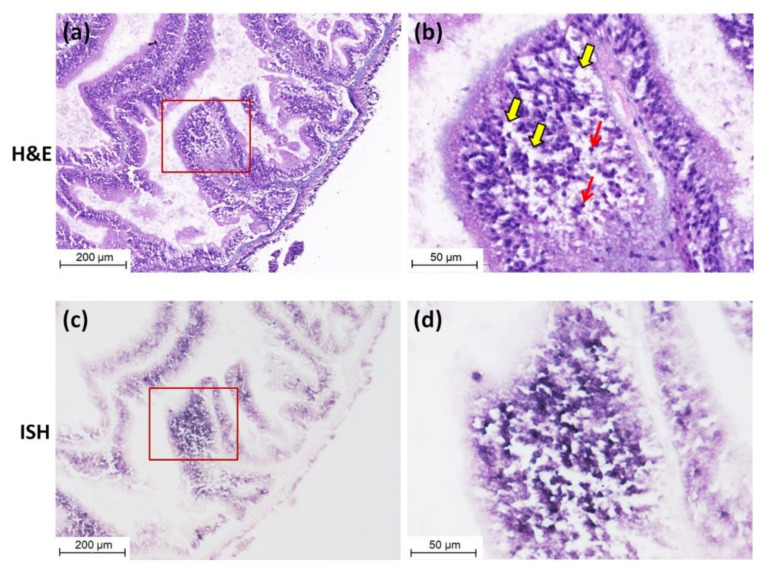
Micrographs of hematoxylin and eosin-phloxine (H&E) staining and *in situ* hybridization (ISH) of the intestine of sea cucumber naturally infected with CMNV. (**a**) Micrographs of H&E staining of the intestine. (**b**) Magnified micrographs from the red-framed areas of (**a**). Karyopyknosis (red arrows) and extensive vacuolation (yellow arrows) were observed in the intestinal epithelial cells. (**c**) Micrographs of ISH of the intestine. (**d**) Magnified micrographs from the red-framed areas of (**c**). Intense CMNV positive hybridization signals (colored deep-purple) were detected in the intestinal epithelial cells. Scale bars: (**a**,**c**) 200 µm, (**b**,**d**) 50 µm.

**Figure 4 viruses-13-00636-f004:**
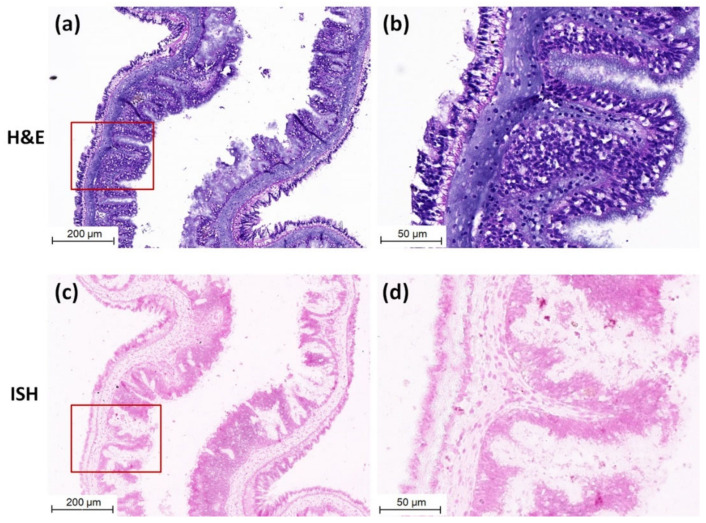
Micrographs of H&E staining and ISH of the intestinal tissue of sea cucumber of negative control. (**a**) Micrographs of H&E staining of negative control intestinal tissue. (**b**) Magnified micrographs from the red-framed areas of (**a**). No obvious histopathological change was observed in the intestinal tissue. (**c**) Micrographs of ISH of negative control intestinal tissue. (**d**) Magnified micrographs from the red-framed areas of (**c**). No CMNV positive hybridization signal was detected in the intestinal tissue. Scale bars: (**a**,**c**) 200 µm, (**b**,**d**) 50 µm.

**Figure 5 viruses-13-00636-f005:**
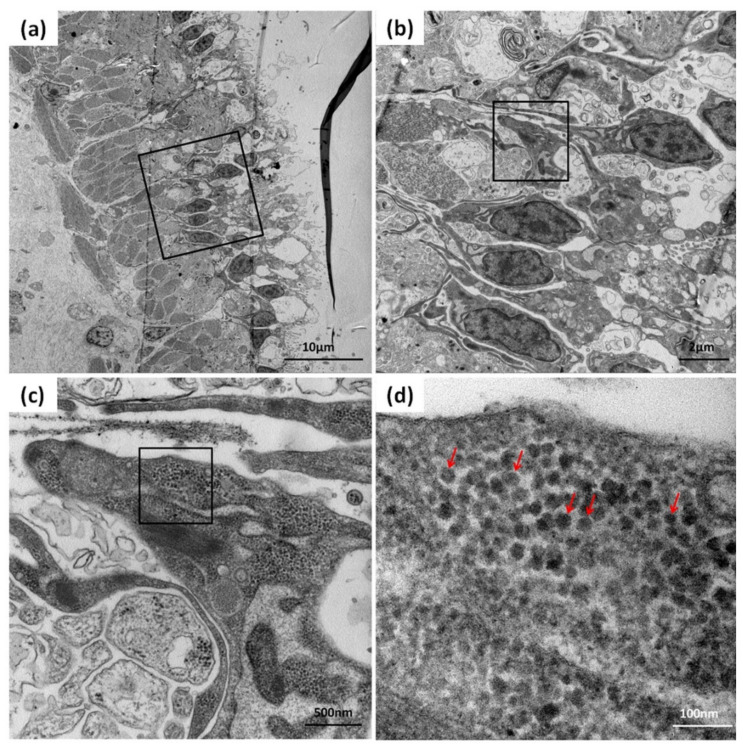
TEM micrographs of an ultrathin section of the intestinal epithelial cells of sea cucumber from the farm infected with CMNV. (**b**–**d**) show magnified micrographs in the black-framed areas of (**a**–**c**), respectively. Note the scattering of CMNV-like virus particles (red arrows) at the intestinal villus epithelium cells. Scale bars: (**a**) 10 µm, (**b**) 2 µm, (**c**) 500 nm, (**d**) 100 nm.

**Table 1 viruses-13-00636-t001:** Names and abbreviations for viral species of Nodaviridae.

Virus	Abbreviation	GenBank No. *
Covert mortality nodavirus	CMNV	AIL48199.1
Flock House virus	FHV	AEQ39075.1
Gungahlin Chrysomya noda-like virus	GCNV	QIJ70031.1
Newington virus	NeV	AMO03244.1
Drosophila melanogaster American nodavirus (ANV) strain SW-2009a	DmANV-SW-2009a	ACU32794.1
Black beetle virus	BBV	YP_053043.1
Wenzhou noda-like virus 6 strain	Wenzhou NLV-6	APG76600.1
Hubei noda-like virus 21 strain	Hubei NLV-21	APG76486.1
Hubei noda-like virus 22 strain	Hubei NLV-22	APG76466.1
Boolarra virus	BoV	NP_689439.1
Nodamura virus	NoV	NP_077730.1
Shuangao insect virus 11 strain	SIV-11	YP_009337806.1
Wenzhou noda-like virus 7 strain	Wenzhou NLV-7	APG76642.1
Beihai noda-like virus 25 strain	Beihai NLV-25	APG76164.1
Beihai mantis shrimp virus 6 strain	BMSV-6	YP_009333376.1
Macrobrachium rosenbergii nodavirus	MrNV	AAQ83832.1
Beihai noda-like virus 30 strain	Beihai NLV-30	APG76125.1
Striped jack nervous necrosis virus	SJNNV	NP_599247
Golden pompano nervous necrosis virus	GPNNV	ACX54065
Redspotted grouper nervous necrosis virus	RGNNV	ACX69744
Dragon grouper nervous necrosis virus	DGNNV	AAU85148
Tiger puffer nervous necrosis virus	TPNNV	YP_00328875
Atlantic cod nodavirus	ACNV	ABR23192
Atlantic halibut nodavirus	AHNV	AAY34458
Barfin flounder nervous necrosis virus	BFNNV	YP_003288756

GenBank No.* indicate the GenBank accession numbers of the amino acid sequence of RNA-dependent RNA polymerase used in this study.

## Data Availability

Not applicable.

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
