# Peer review of "First Report on Natural Infection of Nodavirus in an Echinodermata, Sea Cucumber (Apostichopus japonicas)"

_viruses, 2021, doi:10.3390/v13040636_

Round 1

Reviewer 1 Report

The authors present an interesting study in the detection and observation of Nodavirus in an Echinodermata, sea cucumber (Apostichopus japonicas). This appears to be the first report on Nodavirus infection in this species.  This work is very important because understanding how viruses can infect different species and cross-species barrier is important not only for aquaculture but for human health. Thus, this study sheds light on the broad general ways that Nodaviruses can infect different species. The authors use the methods of  RT-nPCR, histopathology (e.g. H&E staining) and electron microscopy to show nodavirus infection of cells from the sea cucumber. The methods are robust and conclusions justified. This is a good study for publication.

Reviewer 2 Report

It's an interesting and valuable study revealing a new host of the Covert mortality nodavirus. Some comments:

  1. Lanes 71-73. Is still not clear how did you collect sea cucumber samples for RT-nPCR analysis. Did you choose sea cucumber samples just randomly? How many sea cucumber samples have you tested? I wonder also, did these infected sea cucumbers have some symptoms, some differences from healthy individuals?
  2. Lanes 73-73. I don't know if I understood correctly, but these samples were collected from two different sources? One from a previous study in 2018 (lane 71) and another one from this study? 
  3. Lane 117. Table 1. If you have constructed the phylogenetic tree based on the alignment of RdRp gene sequences, why in this table you provide accession numbers of proteins, and not of their genes? 
  4. Is it enough to sequence the fragment of only 413 bp in length to identify a virus, and to be sure what this is truly the same CMNV, that infects also shrimps?
